# Looking for the Edge of the World: How 3D Immersive Audio Produces a Shift from an Internalised Inner Voice to Unsymbolised Affect-Driven Ways of Thinking and Heightened Sensory Awareness

**DOI:** 10.3390/bs13100858

**Published:** 2023-10-19

**Authors:** Sadia Sadia, Claus-Christian Carbon

**Affiliations:** 1School of Art, College of Design and Social Context, RMIT Royal Melbourne Institute of Technology University, Melbourne, VIC 3000, Australia; 2The Light Room, Real World Studios, Wiltshire SN13 8PL, UK; 3Research Group EPÆG (Ergonomics, Psychological Aesthetics, Gestalt), 96047 Bamberg, Bavaria, Germany; 4Department of General Psychology and Methodology, University of Bamberg, 96047 Bamberg, Bavaria, Germany

**Keywords:** psychoacoustics, neuroaesthetics, Dolby Atmos, installation art, sonic arts, inner speech, spatial audio, aesthetic appeal, Gestalt

## Abstract

In this practice-based case study, we investigate the subjective aesthetic and affective responses to a shift from 2D stereo-based modelling to 3D object-based Dolby Atmos in an audio installation artwork. Dolby Atmos is an infinite object-based audio format released in 2012 but only recently incorporated into more public-facing formats. Our analysis focuses on the artist Sadia Sadia’s 30-channel audio installation ‘Notes to an Unknown Lover’, based on her book of free verse poetry of the same title, which was rebuilt and reformatted in a Dolby Atmos specified studio. We examine what effect altered spatiality with an infinite number of ‘placements’ has on the psychoacoustic and neuroaesthetic response to the text. The effectiveness of three-dimensional (3D) object-based audio is interrogated against more traditional stereo and two-dimensional (2D) formats regarding the expression and communication of emotion and what effect altered spatiality with an infinite number of placements has on the psychoacoustic and neuroaesthetic response to the text. We provide a unique examination of the consequences of a shift from 2D to wholly encompassing object-based audio in a text-based artist’s audio installation work. These findings may also have promising applications for health and well-being issues.

## 1. Introduction

‘We travelled on an unnamed roadTo an undiscovered placeLike some sixteenth century explorersLooking for the edge of the world’—from ‘Notes to an Unknown Lover’ (Sadia, 2013)

While immersive audio has been studied for some time, much of that research relates to the film, television, music and gaming industries, which play a large part in driving the economics of audio research [1]. Designing spatial and 360-degree audio for an immersive media experience (IME) ‘might still be considered to be in its infancy’ ([2], p. 90) a nascent field with ‘less defined methods’ ([2], p. 74), many aspects of which are contingent upon the professional practitioners’ subjective evaluations. While any number of studies have been conducted on the effects and application in respect of the moving image as well as popular music [3], little research can be found relating to its uses within the installation of fine arts and even less within the discipline of text-based audio sound design and installation practice.

Sound art as an installation practice and the treatment of sound as a fine art material was arguably first defined by the artist Max Neuhaus in the 1970s [4]. It gained legitimacy and relevance through a number of high-profile exhibitions, including *Sound Art* at the Museum of Modern Art in New York (MoMA, 1979) through to *Sonic Boom: The Art of Sound* at the Hayward Gallery at the Southbank Centre in London (Hayward Gallery, 2000) and the number of artist practitioners and institutional exhibitions continues to grow [5]. It is not our intention to provide a survey of the audio arts in this paper but rather to point to the evolution of sonic practice-based investigation as inextricably linked with the acceleration of audio technologies. This study and the sound art on which it is based would not have been possible without the retail implementation of Dolby Atmos, which drove commercial studios to invest in the technology, making it more readily available to professionals worldwide. The acceleration and accessibility of this technology is a defining factor in our case study as it provides the means and resources on which further research into affect and object-based spatial audio might be based.

For the present practice-based case study, we discuss how 3D immersive audio produces a shift from an internalised inner voice to unsymbolised affect-driven ways of thinking and heightened sensory awareness. This will be achieved by analysing subjective experiences with a 2D text-based audio installation that was reconfigured and transformed into 3D spatial audio by means of a Dolby Atmos-specified studio.

As differentiated from channel-based immersive formats such as binaural and ambisonic, Dolby Atmos is an object-based audio format that is both scalable and flexible, which serves to provide ‘a higher spatial resolution and artistic intent all the way to the playback endpoint’ ([6], p. 244). Originally released in 2012, it has only recently been incorporated into more popular formats by multinationals such as Apple, who introduced the Dolby Atmos format to Apple Music in 2021. In this study, affect is investigated through the use of Descriptive Experience Sampling (DES) to determine whether there is a material alteration in the psychoacoustic perception of the work.

## 2. Materials and Methods

This article is a practice-based case study incorporating research and as such we are defining it under the conventions of arts practice-based research (PBR) case studies [7,8]. The research is based on embodied experience within a highly specialised configuration of materials and equipment uniquely designed for this investigation and concrete artistic material was employed. We are investigating phenomena ‘with a focus on the inner experience and discovery of the researcher’ ([7], p. 113). The intent has been to design a profound experience for an audience of one [9] which could then be extrapolated to a larger group or a wider audience. The process will be described in detail to provide an understanding of how the whole procedure worked. Additionally, we have uploaded all audio-visual material to the OSF platform, see Appendix A (https://osf.io/9ugjq/ (accessed on 11 October 2023)).

### 2.1. Material and Stimuli

This inquiry focuses on an analysis of the creative work ‘Notes to an Unknown Lover’ [10] a thirty-channel audio installation based on the artist’s semi-autobiographical book of poems of the same name, conceived as a reply and homage to Pablo Neruda’s ‘Twenty Love Poems and a Song of Despair’. The artist combines a background as a commercial music record producer with a later career as an installation artist focusing on the moving image and sound design alongside pursuits as an academic research artist with a practice incorporating neuroaesthetics.

Across a cycle of thirty free verse poems, a woman addresses her innermost thoughts to an unidentified and unknown listener. The audio installation comprises thirty discrete speakers, each powered individually and charged overnight. These fully charged speakers were then loaded with a unique micro-SD card containing a single poem as recorded by the artist. Each line of these recorded poems is separated by a fixed period of silence.

When played simultaneously the varying lengths of the lines of text broken by the intervals of silence combine to form a cascade of sound. This also produces a random reading and rewriting of the text, as phrases arise to fill the gaps. As these silences are left vacant or filled with random phrases from other active speakers, unlimited new combinations of the text arise, as the work ‘rewrites’ and reconfigures itself in an infinite number of possible permutations and sequences.

In the original version [10], see Figure 1, the thirty speakers, each representing an individual poem and identified as a unique channel, were mounted on a flat surface in an array against a single wall (see Figure 1). In the reconfigured work [11] produced for this practice-based case study, Dolby Atmos was employed, comprising eleven speaker channels with an infinite number of placements constructed via the algorithmic triangulation of sound. The user defines the placement of the sound and the software instructs it, creating phantom imaging and the psychoacoustic perception of three-dimensional sound location whilst never necessarily coming out of an individual channel. This work was selected for the case study as it provided a confluence of factors, including poetic language-based ‘tracks’ featuring the isolated human voice as well as thirty unique discrete channels which could be repositioned as multiple points of content within an object-based spatial audio system.

### 2.2. Apparatus

Repositioning the original thirty channels of spoken-word text-based audio to 3D object-based surround sound took place in the Dolby Atmos-specified facilities (‘The Red Room’) at Real World Studios, a UK-based studio founded by the musician Peter Gabriel, over a number of sessions held during August 2023. The Red Room was designed by White Mark, whose projects include the London studios of ARD Germany as well as Molinare UK, a leading UK post-production facility. It is equipped with ATC Audio (Glos., UK) monitoring comprising SCM45s for Left, Centre and Right, SCM12s for the sides and tops and an SCM 0.1-15 Pro Subwoofer [12] (see Figure 2).

Dolby Atmos speaker configuration in basic format is 7.1.4 with seven speakers located left front, centre, right front, left side, right side, left rear, right rear, one LFE low-frequency effect subwoofer at ground level and four speakers located left front height, right front height, left rear height and right rear height. We used our own Digital Audio Workstation (DAW) running Pro Tools Ultimate with Dolby Atmos Renderer and our systems integrated seamlessly with the control room hardware. Pro Tools allows for the separation of each audio channel to be defined as an ‘object’ which can be positioned any-where within the 3D space. These objects become audible and visible within the Dolby Atmos Renderer, which also displays the approximate levels sent to each of the available speaker channels.

### 2.3. Procedure

The original 2D version of the NTAUL stimuli/input audio was processed in 3D Dolby Atmos with playback taking place in The Red Room (see Figure 2) at Real World Studios across two sessions on 3 August 2023 and 11 August 2023.

Methodologically observations described herein are based on informed subjective interpretation as an investigative method in the analysis of the experience of contemporary sound design [13] alongside an adapted form of Descriptive Experience Sampling (DES) as a protocol for accessing pristine inner experience, limiting investigation to ‘specific, clearly identified moments’ and ‘directly apprehended (‘pristine’) inner experience’ [14]. A mobile phone alarm app was set to emit a prompt at set intervals during which the researcher recorded episodic representations of the inner mental state. Results from the 3 August 2023 session, which incorporated the initial design phase and object placement, were informally recorded. Results from the 11 August 2023 session were structured according to DES sampling methods employing episodic prompts and can be found in Table 1.

DES is increasingly used across a wide range of disciplines including music, well-being and the phenomenological investigation of the music festival experience (Moss, Whalley, Ellesmore, 2020). It was selected over other experience sampling methods (ESM) as particularly applicable to audio, where perception may be fleeting. While ‘typical sampling methods and questionnaires may appear to investigate the phenomena of pristine inner experience, that appearance is largely illusory. Such studies should be thought of not as investigations of pristine experience, but rather as investigations of some ill-defined mixture of presuppositions or judgments about experience and pristine experience itself’ ([14], p. 148). DES ‘allows for both qualitative and quantitative data to be collected, to measure specific elements of experience or to seek out those as yet unidentified’ ([15], p. 388) and is well suited to sonic arts and audio applications.

The following sequence of three audio representations were run (all materials can be played on the OSF site as described above in the Section 2):(1)Sadia, S. (2014) [10]. Notes to an Unknown Lover. Fine art sound installation.

Original: Audio representation of the original 2D NTAUL audio installation. To be listened to on stereo speakers only (not headphones). Please see:

Appendix A (https://osf.io/9ugjq/ (accessed on 11 October 2023))

(2)Sadia, S. (2023) [11]. Notes to an Unknown Lover V.2. Fine art sound installation.

Transition: Demonstration of Dolby Atmos Renderer illustrating the transition to 3D Dolby Atmos repositioning. Audio is presented in binaural to be listened to on headphones or earbuds only (to approximate the 3D listening space). Please see:

Appendix A (https://osf.io/9ugjq/ (accessed on 11 October 2023))

(3)Sadia, S. (2023) [11]. ‘Notes to an Unknown Lover’ V.2. Fine art sound installation.

Example: Video sample of 3D NTAUL Dolby Atmos audio installation with repositioning. Audio is presented in binaural to be listened to on headphones or earbuds only (to approximate the 3D listening space). Please see:

Appendix A (https://osf.io/9ugjq/ (accessed on 11 October 2023))

## 3. Results and Discussion

When approaching a stereo or 2D modelled audio installation, the viewer will cross liminal space, the threshold over which they approach the artwork. The poem included in the introduction speaks to that liminal space and the challenges inherent in adapting creative works to more effectively impact an audience. Burickson, LeRoux and Moody ([9], p. 8) referred to this process as framed by the question ‘How could our art have the deepest possible impact on our audience?’. Artist practitioners who engage with neuroaesthetics are concerned not only with how phenomenal experience may be processed by their participants or audiences but also with how they might adapt their work to achieve the most profound, transcendent or heightened affective response to their creative outputs.

In the action of crossing this liminal space, the viewer or participant exercises agency in that they may move towards or away from the 2D modelled audio source at will, allowing for an element of autonomous, active control over the embodied experience. On approach, volume may increase and the sense of being immersed within the work will also increase with proximity. Critical to this observation is the identification of ‘self’ and ‘other’ as the boundary between inner experience (‘self’ or ‘I’) and external stimuli (‘other’) is mediated through the participant’s agency. Sound is processed through the movement of the head and the waveforms strike the body but the skin receives the audio through ventral surfaces only. The reception of the embodied experience is therefore limited, controllable and mensurable by the participant.

The case for 3D object-based audio requires a surrender of these elements of agency. In respect of Dolby Atmos, the body is wholly surrounded but only from the waist up (depending on whether the subject is sitting or standing) and speakers are located at seated level and above but not below. Sound strikes both the dorsal and ventral planes of the body and while perspective and relative balances can be materially altered by moving about the exhibition or studio space, it is impossible to alter the relative level by moving ‘away’ from the work as there exists no means of withdrawing from the work other than by exiting the space completely. The 2D audio installation is, therefore, relativistic in that a viewer or listener may exercise agency and alter the intensity of the experience by positioning themselves at a distance in space, while the 3D audio environment is binary in that the participant is either in the space experience or wholly outside it (a parallel might be drawn to the difference between a dimmer switch and an on/off switch).

Music perception and its related cognitive processes have been compared with language processing and understanding [16] and this is the space in which text-based audio installation and affect-driven response to music meet. The work invokes the pragmatist philosophy of John Dewey, with perception as a communicative process that connects both the artist and the perceiver [17] to their shared environment through memory and emotion [18] with ‘emotion as the moving and cementing force’ ([19], p. 9).

Spoken word can share characteristics with music in pitch, timbre and tuning alongside the effects of voice on emotional arousal [20]. The perception is of a shift from an externalised voice (2D) to the voice inside your head (3D) [21], supplanting the inner monologue or dialogue in those subjects who experience inner speech, an ‘introspectively salient component of everyday experience’ ([22], p. 2). While not many have a continuous inner monologue, most people experience inner speech as a monologic or dialogic conversation some of the time [23] or most of the time [24]. The binary on/off nature of the 3D spatial audio experience combined with the concomitant loss of control and the pervasive nature of the media allows the work to supplant inner speech or dialogue. The effect of the shift from two-dimensional to three-dimensional spatial audio in ‘Notes to an Unknown Lover’ was to still the inner monologue/dialogue, replacing it with the artwork’s recorded text as assimilated by the body from thirty surrounded object points (see Figure 3). Replacing the inner voice with the voices from the audio artwork produces a shift from internalised inner voice to unsymbolised affect-driven ways of thinking and heightened sensory awareness. The emotional information contained in the spoken word is affected by the alteration in perception brought about by the shift from 2D spatial modelling to 3D wholly encompassing audio modelling, with the cognitive shift brought about through the altered localisation of sound. The stilling or replacing of inner speech has applications among those who suffer from negative internal dialogue as inner speech is significantly correlated with states of anxiety, depression and other psychopathologies [25]. This indicates a symbiotic relationship between the disciplines of spatial audio technologies, neuroaesthetics and the therapeutic arts whose potential value makes a case for further investment and research.

**Figure 3 behavsci-13-00858-f003:**
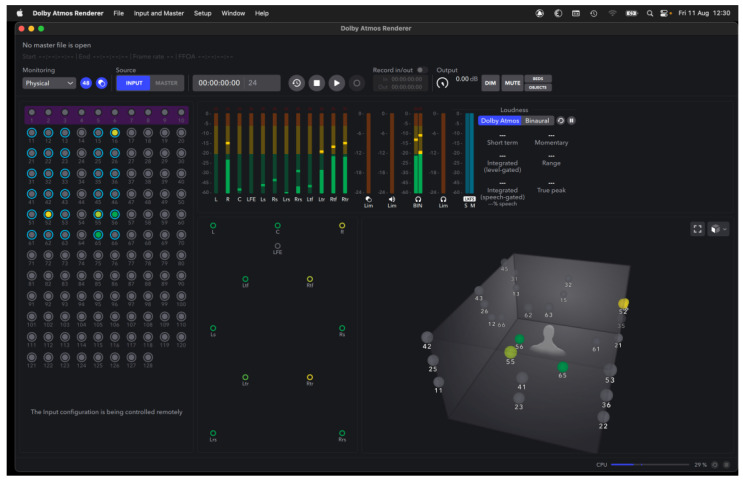
Sample of object placement of NTAUL repositioned audio in Dolby Atmos Renderer.

**Table 1 behavsci-13-00858-t001:** Descriptive Experience Sampling (DES) of repositioned thirty-channel Dolby Atmos audio installation taken at two-minute intervals. Session 2, The Red Room, Real World Studios, 11 August 2023.

Time	DES Descriptive Experience Sampling (‘Pristine’) NTAUL|30-Channel 3d Dolby Atmos
1 min	Inner dialogic voice becomes monologic.
3 min	Monologic voice quiets; becomes a visualisation.
5 min	Becomes conscious of breath and the action of breathing; chest rising and falling.
7 min	3D audio replaces inner monologue; sound is ‘internalised’.
9 min	Disembodiment; a sense of floating in limitless space; walls of room ‘disappear’.
11 min	Loss of sense of ‘self’ or ‘I’; replaced by audio; disembodiment.
13 min	Reported meditative state; dissociative.
15 min	Disembodied state; sound moves ‘through’ ‘inside’ and ‘out’.
17 min	Stilled internal monologue replaced by embodied audio; a sense of peace.
19 min	Mind becomes conscious of emotional content of spoken word; emotional.

In Session (2) (11 August 2023), the audio installation NTAUL (2014) was fully repositioned in 3D with the subject sitting in the optimum location seated in the centre of the Atmos studio and the full 30-channel spectrum allowed to run in Dolby Atmos. Descriptive Experience Sampling (DES) was employed to access pristine inner experience at two-minute intervals (see Table 1).

While the 2D modelling of the audio installation resulted in no interruption of the ‘inner voice’, the 3D Dolby Atmos repositioned work resulted in a complete stilling of both monologic and dialogic inner speech. This returns us to pragmatist aesthetics in opening the door to Dewey’s emotion as the ‘moving and cementing force’ (Dewey, 1934, [19] 2005), the stilled inner voice opening space for embodied experience and an internalised integration with the encompassing media. The work NTAUL (2014), on which this practice-based case study is based, was originally intended to be a wholly encompassing sonic art installation. In 2014, object-based spatial audio systems were not readily available outside of highly specialised cinema audio applications. A seismic historical shift occurred with the adoption of Dolby Atmos by companies such as Apple in 2021. Accessibility of technology drives innovation and, in this instance, allowed for the reconfiguration of a 2D audio installation to 3D. On a cognitive and neuroaesthetic level, the deployment of this technology enhances the psychoacoustic perception of NTAUL and more accurately reflects the artist’s intention.

## 4. Conclusions

The shift from 2D audio modelling to 3D Dolby Atmos is accompanied by a concomitant loss of agency by the participant in the negotiation of the experience. This loss of control is accompanied by heightened arousal and responsiveness to the emotional information contained in the spoken word in the 3D environment. While stereo and 2D do not appear to silence monologic or dialogic inner voices, thought and emotion sampling procedures indicate that 3D object-based Dolby Atmos surrounded audio sound design may be effective in stilling these voices and in influencing the inner experience state. These findings may have promising applications for health and well-being issues in particular. Additional research is indicated into the study of affect, anxiety and depression, as well as neurodiversity, in the context of and in relation to object-based spatial audio environments. This calls for further, systematic research across cultures and demographics, always taking place in an adequate context to allow the artistic experience to happen [26].

## Figures and Tables

**Figure 1 behavsci-13-00858-f001:**
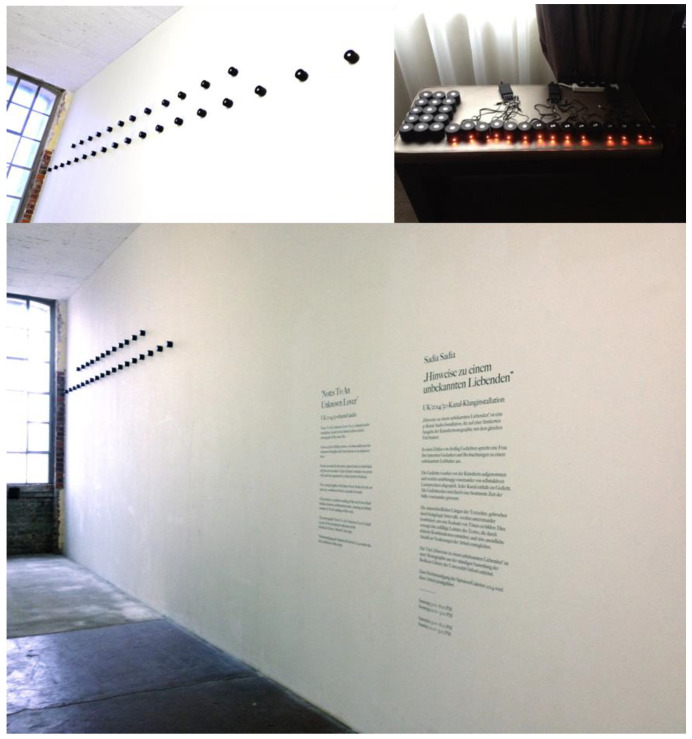
Sadia, S. (2014) [10]. Notes to an Unknown Lover. Thirty-channel audio installation. Speakers, micro-SIM cards. Spinnerei Rundgang, Leipzig 2014. A version of this installation was exhibited in ‘Poetry’, George Paton Gallery, University of Melbourne in 2017.

**Figure 2 behavsci-13-00858-f002:**
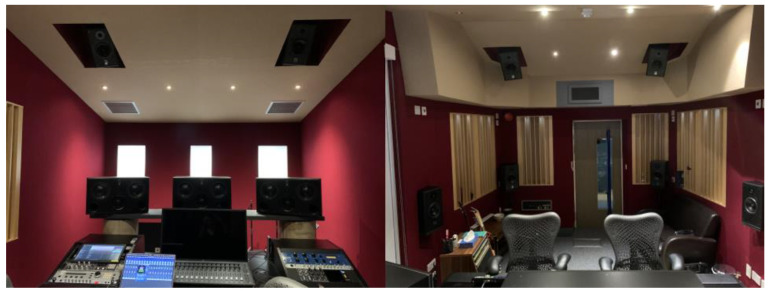
The Red Room, Dolby Atmos speaker placement. Real World Studios, August 2023.

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
