# Peer review of "Looking for the Edge of the World: How 3D Immersive Audio Produces a Shift from an Internalised Inner Voice to Unsymbolised Affect-Driven Ways of Thinking and Heightened Sensory Awareness"

_behavsci, 2023, doi:10.3390/bs13100858_

Round 1

Reviewer 1 Report

This was a very compelling paper. It is very clear in its aims and carefully describes the experiment the authors developed but there are three big questions I have for the authors and I think the paper would be strengthened if they could address them therein.

1. The authors describe this as an "opinion piece" but it seems to me that they conducted an experiment and are reporting on the results. Could they explain why they see this as an opinion rather than a study? How does this affect the results they present?

2. The authors say on line 213 that the second version of the work more accurately reflects the artist's intention. Seeing that the artist is one of the authors, I understand the validity of this claim, but I still want to know more about why this manifestation of the work is better in terms of the artist's intention. What aspects of her intention are relevant in this assertion?

3. DES descriptive experience sampling--What is this exactly and how does it work? I think that Table 1. is really useful as a descriptor but I want to know the process of sampling, the number of people who responded to the sampling (how big is the group if individuals who were measured? Do all of them have the same response? Or are there just the two authors doing the DES?) If the last is true, that would make sense why this is an opinion piece, but the process of sampling needs to be explained.

Author Response

Reviewer One

This was a very compelling paper.

> Thank you.

It is very clear in its aims and carefully describes the experiment the authors developed but there are three big questions I have for the authors and I think the paper would be strengthened if they could address them therein.

  1. The authors describe this as an "opinion piece" but it seems to me that they conducted an experiment and are reporting on the results. Could they explain why they see this as an opinion rather than a study? How does this affect the results they present?

> At the suggestion of Reviewer (2) the paper has been reframed as a practice-based case study

  1. The authors say on line 213 that the second version of the work more accurately reflects the artist's intention. Seeing that the artist is one of the authors, I understand the validity of this claim, but I still want to know more about why this manifestation of the work is better in terms of the artist's intention. What aspects of her intention are relevant in this assertion?

>please see lines 298-310

The work NTAUL (2014), on which this practice-based case study is based, was originally intended to be a wholly encompassing sonic art installation. In 2014, object-based spatial audio systems were not readily available outside of highly specialized cinema audio applications. A seismic historical shift occurred with the adoption of Dolby Atmos by companies such as Apple in 2021. Accessibility of technology drives innovation and in this instance allowed for the reconfiguration of a 2D audio installation to 3D. On a cognitive and neuroaesthetic level, the deployment of this technology enhances the psychoacoustic perception of NTAUL and more accurately reflects the artist’s intention.”

  1. DES descriptive experience sampling--What is this exactly and how does it work? I think that Table 1. is really useful as a descriptor but I want to know the process of sampling, the number of people who responded to the sampling (how big is the group if individuals who were measured? Do all of them have the same response? Or are there just the two authors doing the DES?) If the last is true, that would make sense why this is an opinion piece, but the process of sampling needs to be explained.

>please see lines 80-89

This article is a practice-based case study incorporating research and as such we are defining it under the conventions of arts practice-based research (PBR) case studies (McNiff, 1998; Candy, 2006). The research is based on embodied experience within a highly specialised configuration of materials and equipment uniquely designed for this investigation and concrete artistic material was employed. We are investigating phenomena “with a focus on the inner experience and discovery of the researcher” (McNiff, 1998, pg 113). The intent has been to design an effective experience for an audience of one (Burickson, Leroux, 2016) which could then be extrapolated to a larger group or a wider audience.”

>please see lines 174-190

A mobile phone alarm app was set to emit a prompt at set intervals during which the researcher recorded episodic representations of the inner mental state. Results from the August 3rd 2023 session, which incorporated initial design and object placement, were informally recorded. Results from the August 11th 2023 session were structured according to DES sampling methods employing episodic prompts and can be found in Table 1.

“DES is increasingly used across a wide range of disciplines including music and wellbeing through to experience sampling of music festival experience (Moss, Whalley, Ellesmore, 2020). It was selected over other Experience Sampling Methods (ESM) as particularly applicable to audio where perception may be fleeting. While “typical sampling methods and questionnaires may appear to investigate the phenomena of pristine inner experience, that appearance is largely illusory. Such studies should be thought of not as investigations of pristine experience, but rather as investigations of some ill-defined mixture of presuppositions or judgments about experience and pristine experience itself” (Hurlburt, Heavey, 2015, pgs 148-149). DES “allows for both qualitative and quantitative data to be collected, to measure specific elements of experience or to seek out those as yet unidentified”(Moss, Whalley, Elsmore, 2020, p 388) ) and is well suited to sonic arts and audio applications.”

English language fine. No issues detected

Reviewer 2 Report

This is an interesting and innovative project and I think it is important to interrogate fine art practices in order to discover what we know about the world and our diverse experiences in it. We have different types of knowing and this project seeks to explore them.    I think this is a practice-based case study rather than opinion piece because it is reporting on an experiment and reflecting on the results, rather than discussing an issue. If it was a longer paper I would recommend it being positioned as a piece of practice-based research.

I would introduce the artist  Sadia, S. and the poem in the introduction and explain why their work has been used as a case study.  I would also look at current fine art installation practice that includes sound and so the authors can position their case study in relation to that.

I would also reflect on the ethics of this project if it was opened up to an audience, how would  the authors would  safe guard against the experience being difficult for some people who have different sensitivities to sound or may be neuro-diverse. Also reflect that an embodied experience may be different for everyone, if someone who could not hear entered this exhibition could they access it through its visuality, haptic qualities and its vibration?  

I would like to know how the table on lines 200-201 was created? How was the experience measured/observed who experienced these effects? If it was the artist or the producer of the piece then that is fine, but some transparency about this would be welcomed.

The findings suggest that the internal voice is stilled by the experience, why would this be beneficial? Why is linked to wellbeing?

I also noted the links and passwords in the text, I would append these at the end of the text, so the flow of the writing is not interrupted, and think about making it open access rather than passwords, maybe the editors of the journal can give further guidance on this.

A good copyedit check would benefit the paper, looking for word order and missing words.

There was a lot of technical information and I noted a short glossary was included.  I think to be consistent and methodical in always spelling out the  name in full then including the acronym in brackets afterwards when first introducing it would be best for the reader: eg Descriptive Experience Sampling (DES)

Author Response

Reviewer Two

This is an interesting and innovative project and I think it is important to interrogate fine art practices in order to discover what we know about the world and our diverse experiences in it.

> Thank you very much for your positive overall evaluation!

1) We have different types of knowing and this project seeks to explore them.   I think this is a practice-based case study rather than opinion piece because it is reporting on an experiment and reflecting on the results, rather than discussing an issue. If it was a longer paper I would recommend it being positioned as a piece of practice-based research.

> Many thanks for this observation. Great suggestion. The paper has been reframed as a practice-based case study. This is indeed a better fit.

2) I would introduce the artist  Sadia, S. and the poem in the introduction and explain why their work has been used as a case study.  

>please see lines 95-98

The artist combines a background as commercial music record producer with a later career development as an installation artist focusing on the moving image and sound design alongside pursuits as an academic research artist with a practice incorporating neuroaesthetics.”

>please see lines 133-137

This work was selected for the case study as it provided a confluence of factors, including poetic language-based ‘tracks’ featuring the isolated human voice as well as thirty unique discrete channels which could be repositioned as multiple points of content within an object-based spatial audio system.”

>please see lines 222-230

The poem included in the introduction speaks to that liminal space and the challenges inherent in adapting creative works to more effectively impact an audience. Burickson and Leroux (2016) refer to this process as framed by the question “How could our art have the deepest possible impact on our audience?”(ibid, p 8). Artist practitioners who engage with neuroaesthetics are concerned not only with how phenomenal experience may be processed by their participants or audiences but also how they might adapt their work to achieve the most profound, transcendent or heightened affective response to their creative outputs.”

3) I would also look at current fine art installation practice that includes sound and so the authors can position their case study in relation to that.

>please see lines 52-65

Sound art as an installation practice and the treatment of sound as a fine art material was arguably first defined by the artist Max Neuhaus in the 1970’s (Dunaway, 2020). ). It gained legitimacy and relevance through a number of high-profile exhibitions, including Sound Art at the Museum of Modern Art in New York (MoMA, 1979) through to Sonic Boom: The Art of Sound at the Hayward Gallery at the Southbank Centre in London (Hayward Gallery, 2000) and the number of artist practitioners and institutional exhibitions continues to grow (Downey, 2022). It is not our intention to provide a survey of the audio arts in this paper, but rather to point to the evolution of sonic practice-based investigation as inextricably linked with the acceleration of audio technologies. This study and the sound art on which it is based would not have been possible without the retail implementation of Dolby Atmos, which drove commercial studios to invest in the technology making it more readily available to professionals worldwide. The acceleration and accessibility of this technology is a defining factor in our case study as it provides the means and resources on which further research into affect and object-based spatial audio might be based.”

4) I would also reflect on the ethics of this project if it was opened up to an audience, how would  the authors would safe guard against the experience being difficult for some people who have different sensitivities to sound or may be neuro-diverse.

Affect and spatial is a nascent field of research with contributions largely relating to commercial applications. No research that we can identify has been done into deafness or neurodiversity in response to object-based spatial audio. We hesitate to speculate until such research is conducted under the appropriate ethics committee approvals. There is now a call for further research included in the paper (as indicated on lines 321-323 below).

>Please also see Lines 280-283

This indicates a symbiotic relationship between the disciplines of spatial audio technologies, neuroaesthetics and the therapeutic arts whose potential value makes a case for further investment and research”

>please see lines 321-323

Additional research is indicated into the study of affect, anxiety and depression, as well as neurodiversity, in the context of and in relation to object-based spatial audio environments.”

5) Also reflect that an embodied experience may be different for everyone, if someone who could not hear entered this exhibition could they access it through its visuality, haptic qualities and its vibration?  

Very low vibrations might be perceived but not higher frequencies. Please see reply to (4).

6) I would like to know how the table on lines 200-201 was created? How was the experience measured/observed who experienced these effects? If it was the artist or the producer of the piece then that is fine, but some transparency about this would be welcomed.

>please see lines 80-88

This article is a practice-based case study incorporating research and as such we are defining it under the conventions of arts practice-based research (PBR) case studies (McNiff, 1998; Candy, 2006). The research is based on embodied experience within a highly specialised configuration of materials and equipment uniquely designed for this investigation and concrete artistic material was employed. We are investigating phenomena “with a focus on the inner experience and discovery of the researcher” (McNiff, 1998, pg 113). The intent has been to design an effective experience for an audience of one (Burickson, Leroux, 2016) which could then be extrapolated to a larger group or a wider audience.”

> please see lines 174-190

A device was set to emit a prompt at set intervals during which the researcher recorded episodic representations of inner state. Results from the August 3rd 2023 session, which incorporated initial design and object placement, were informally recorded. Results from the August 11th 2023 session were structured according to DES sampling methods employing episodic prompts and can be found in Table 1.

“DES is increasingly used across a wide range of disciplines including music and wellbeing through to experience sampling of music festival experience (Moss, Whalley, Ellesmore, 2020). It was selected over other Experience Sampling Methods (ESM) as particularly applicable to audio where perception may be fleeting and while “typical sampling methods and questionnaires may appear to investigate the phenomena of pristine inner experience, that appearance is largely illusory. Such studies should be thought of not as investigations of pristine experience, but rather as investigations of some ill-defined mixture of presuppositions or judgments about experience and pristine experience itself” (Hurlburt, Heavey, 2015, pgs 148-149). DES “allows for both qualitative and quantitative data to be collected, to measure specific elements of experience or to seek out those as yet unidentified”(Moss, Whalley, Elsmore, 2020, p 388) ) and is well suited to sonic arts and audio applications”.

7) The findings suggest that the internal voice is stilled by the experience, why would this be beneficial? Why is linked to wellbeing?

>please see lines 278-283

The stilling or replacing of inner speech has applications among those who suffer from negative internal dialogue as inner speech is significantly correlated with states of anxiety, depression and other psychopathologies (Alderson-Day, et al, 2018). This indicates a symbiotic relationship between the disciplines of spatial audio technologies, neuroaesthetics and the therapeutic arts whose potential value makes a case for further investment and research”.

>please see lines 321-323

Additional research is indicated into the study of affect, anxiety and depression, as well as neurodiversity, in the context of and in relation to object-based spatial audio environments.”

8) I also noted the links and passwords in the text, I would append these at the end of the text, so the flow of the writing is not interrupted, and think about making it open access rather than passwords, maybe the editors of the journal can give further guidance on this.

>The audio-visual files provide important data on the procedure. The links with passwords are a temporary measure for review only. The files should be hosted through MDPI on publication according to their protocols and will be open access. They carry the CC Creative Commons copyright notice for open access as required by MDPI on the end screen.

At the moment, the AV files are also listed under ‘Supplementary materials’ at the end of the document.

I believe this will be addressed during the final production phase. We will keep an eye on it to ensure that all material is made available.

Comments on the Quality of English Language

9) A good copyedit check would benefit the paper, looking for word order and missing words.

>The document has been copy edited and tidied up. Duplications have been deleted. Spelling, grammar and capitalisation have been addressed. The chosen convention in this document is to favour British English (for example, ‘s’ over ‘z’ and ‘-our’ over ‘-or’) over American English. The Oxford comma has been employed for clarity as appropriate.

10) There was a lot of technical information and I noted a short glossary was included.  I think to be consistent and methodical in always spelling out the name in full then including the acronym in brackets afterwards when first introducing it would be best for the reader: eg Descriptive Experience Sampling (DES).

>Yes, many thanks. Updated in document with name in full then acronym in brackets thereafter when first introducing and consistent throughout.

Reviewer 3 Report

This opinion piece describes the experiential differences between a 2-D stereo-based sound modeling and a 3-D object-based Dolby Atmos audio installation for an artwork installation. The paper is well-written, but the structure presents the work as an opinion piece without method and a methods-based approach using a single user (n=1) and introspection. The Descriptive Experience Sampling suggests data was collected from multiple users, but this is not clear from the contents. The paper needs to be either an opinion piece OR participant-based research. 

The use of English is excellent. 

Author Response

Reviewer Three

This opinion piece describes the experiential differences between a 2-D stereo-based sound modeling and a 3-D object-based Dolby Atmos audio installation for an artwork installation.

1) The paper is well-written, but the structure presents the work as an opinion piece without method and a methods-based approach using a single user (n=1) and introspection. The Descriptive Experience Sampling suggests data was collected from multiple users, but this is not clear from the contents.

> please see 80-88

This article is a practice-based case study incorporating research and as such we are defining it under the conventions of arts practice-based research (PBR) case studies (McNiff, 1998; Candy, 2006). The research is based on embodied experience within a highly specialised configuration of materials and equipment uniquely designed for this investigation and concrete artistic material was employed. We are investigating phenomena “with a focus on the inner experience and discovery of the researcher” (McNiff, 1998, pg 113). The intent has been to design an effective experience for an audience of one (Burickson, Leroux, 2016) which could then be extrapolated to a larger group or a wider audience.”

>please see 177-194

“A mobile phone alarm app was set to emit a prompt at set intervals during which the researcher recorded episodic representations of the inner mental state. Results from the August 3rd 2023 session, which incorporated initial design and object placement, were informally recorded. Results from the August 11th 2023 session were structured according to DES sampling methods employing episodic prompts and can be found in Table 1.”

2) The paper needs to be either an opinion piece OR participant-based research. 

> This does indeed make a great deal of sense. At the suggestion of Reviewer (2) the paper has been reframed as a practice-based case study.

3) The use of English is excellent.

> Many thanks.

Round 2

Reviewer 3 Report

This paper is much improved. My only comment regards including the number (n) of participants who participated in the PBR of the work. Otherwise, the methods section describes the two methods used in the evaluation.